# Solvable lattice models for metals with Z2 topological order

**Brin Verheijden[1], Yuhao Zhao[2] and Matthias Punk[3,4*]**

**1** Physics Department, Ludwig-Maximilians-University Munich, D-80333 München
**2** Department of Physics, ETH Zurich, 8093 Zürich, Switzerland
**3** Physics Department, Arnold Sommerfeld Center for Theoretical Physics and Center
for NanoScience, Ludwig-Maximilians-University Munich, D-80333 München
**4** Munich Center for Quantum Science and Technology (MCQST),
Schellingstr. 4, D-80799 München, Germany

⋆ matthias.punk@lmu.de

## Abstract

We present quantum dimer models in two dimensions which realize metallic ground states with Z2 topological order. Our models are generalizations of a dimer model introduced in [PNAS 112, 9552-9557 (2015)] to provide an effective description of unconventional metallic states in hole-doped Mott insulators. We construct exact ground state wave functions in a specific parameter regime and show that the ground state realizes a fractionalized Fermi liquid. Due to the presence of Z2 topological order the Luttinger count is modified and the volume enclosed by the Fermi surface is proportional to the density of doped holes away from half filling. We also comment on possible applications to magic-angle twisted bilayer graphene.

# 1   Introduction

Landau's Fermi liquid theory is one of the cornerstones of condensed matter physics and is remarkably successful in describing conventional metallic phases of interacting electrons. Despite its wide success, various strongly correlated electron materials show unconventional metallic behavior which does not fit into the Fermi liquid framework. One prime example are the cuprate high-$T_c$ superconductors, which exhibit a distinct non-Fermi liquid or "strange metal" phase around optimal doping [1, 2], as well as a metallic "pseudogap" phase at low hole-doping featuring Fermi-liquid like transport properties, but an anomalously low charge carrier concentration [3–5]. Interestingly, phases with a similar non-Fermi liquid phenomenology have been recently observed in magic-angle twisted bilayer graphene [6, 7] .

The theoretical description of unconventional metallic phases in dimensions $d \geq 2$ largely focuses on two broad classes of non-Fermi liquids. The first are metals without well-defined electronic quasiparticle excitations and appear e.g. in the phenomenological theory of marginal Fermi liquids [8], in the vicinity of metallic quantum critical points [9], or have recently been discussed in the context of SYK models [10, 11]. A second class of models with non-Fermi liquid phenomenology is based on the concepts of topological order and fractionalization, where electronic degrees of freedom fractionalize into partons, each carrying some of the quantum numbers of an electron [12–17]. One striking consequence of topological order in such models is the possibility to modify Luttinger's theorem, which states that the volume enclosed by the Fermi surface is proportional to the density of electrons in the conduction band for ordinary Fermi liquids [18]. By contrast, so-called fractionalized Fermi liquids, which have been introduced originally in the context of heavy Fermion systems [19], provide a prime example for metallic phases with a modified Luttinger count and feature a reconstructed, small Fermi surface in the absence of broken translational symmetries.

In this work we present exctly solvable, two-dimensional lattice models that exhibit metallic ground states with Z2 topological order. These models are defined on the non-bipartite triangular and kagome lattices and are generalizations of a quantum dimer model introduced in Ref. [20]. The latter is defined on the square lattice and has been argued to capture some of the unusual electronic properties of the pseudogap phase in underdoped cuprates and is itself a generalization of the well-known Rokhsar-Kivelson (RK) model [21]. Its Hilbert space is spanned by hard-core configurations of bosonic spin-singlet dimers, as well as fermionic spin-1/2 dimers carrying charge $q = +e$, both living on nearest neighbor bonds. Fermionic dimers represent a hole in a bonding orbital between two neighboring lattice sites and can be viewed as bound states of a spinon and a holon in a doped resonating valence bond liquid. The density of fermionic dimers is equal to the density of doped holes away from half filling and the model reduces to the RK model at half filling. Subsequent numerical studies of the square lattice model computed single-electron spectral functions and revealed the presence of an anti-nodal pseudogap as well as Fermi arc-like features in the electron spectral function [22]. Furthermore, exact ground-state wave functions were constucted for a specific parameter regime in Ref. [23].

Due to the fact that the model in Ref. [20] is defined on the bipartite square lattice and only allows for nearest neighbor dimers, a fractionalized Fermi liquid ground state without broken symmetries only appears by fine-tuning to the special RK-point, where the bosonic dimers form a $U(1)$ spin liquid. This is not a stable phase of matter, however, as the $U(1)$ spin liquid is considered to be confining at large length scales [24, 25]. By contrast, the analogous quantum dimer models on the non-bipartite triangular and kagome lattices constructed in this work feature a stable fractionalized Fermi liquid (FL*) phase and don't require fine-tuning. At half-filling, i.e. at a vanishing density of fermionic dimers, these models exhibit an extended Z2 spin liquid phase [26] and realize a metallic Z2-FL* ground state without broken symmetries

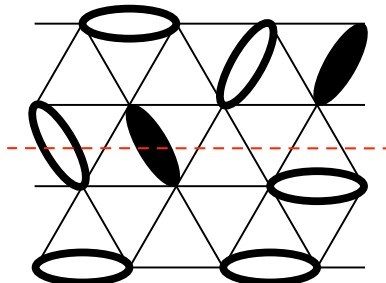

Figure 1: Example of a hard-core dimer configuration spanning the Hilbert space of our triangular lattice model. White ellipses represent bosonic spin-singlet dimers, black ellipses are fermionic dimers. The parity of the number of bosonic and fermionic dimers crossing the red dashed line does not change under arbitrary local dimer rearrangements, as expected for a state with $Z2$ topological order.

upon doping with holes [17].

Lastly we also note that a metallic phase with an unusually low charge carrier density has been observed in magic-angle twisted bilayer graphene (TBG) [6]. It appears upon hole-doping the Mott-like insulating phase at half filling of the lower Moiré mini-band close to charge neutrality, in remarkable analogy with the pseudogap phase in underdoped cuprates. In this work we argue that the triangular lattice quantum dimer model presented here could provide a toy model for the description of this unconventional metallic phase and we point out specific signatures which can be checked in future experiments. In particular we argue that the Fermi surface in the Z2-FL* phase of the triangular lattice dimer model consists of small hole pockets centered at the M points of the Brillouin zone (i.e. the Moiré mini-Brillouin zone in the case of twisted bilayer graphene). Angle resolved photoemission (ARPES) experiments with a sufficiently high momentum resolution or quasi-particle interference in scanning tunnneling microscopy (STM) experiments on TBG should be able to test this prediction.

The rest of the paper is outlined as follows: in Sec. 2 we introduce the two-species quantum dimer model on the triangular lattice and construct exact ground states at a specific line in parameter space in Sec. 3. In Sec. 4 we consider perturbations away from the exactly solvable line and compare the perturbative results with numerical exact diagonalization data. Moreover, we discuss the emergence of a Z2-FL* ground state and compute the single electron spectral function. Finally, in Sec. 5 we discuss possible applications of this model to magic-angle twisted bilayer graphene. In the appendix we briefly present a construction of exact ground states for analogous dimer models on the kagome lattice.

## 2 Triangular lattice dimer model

As in Ref. [20] the Hilbert space of our model is spanned by hard-core coverings of the triangular lattice with two kinds of dimers living on nearest neighbor bonds. A bosonic dimer which represents two electrons in a spin-singlet configuration as in the usual Rokhsar-Kivelson model, as well as a fermionic spin-1/2 dimer carrying electric charge $+e$ with respect to a bosonic dimer background. The latter represents an electron in a bonding orbital delocalized between two neighboring lattice sites and can also be viewed as a tightly bound spinon-holon pair in a doped resonating valence bond liquid. An example of a dimer configuration is shown in Fig. 1.

Dynamics on this Hilbert space is generated by a Hamiltonian including various local dimer

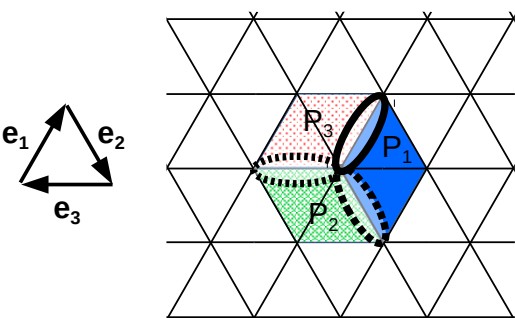

Figure 2: The three elementary plaquettes per lattice site, labelled $P_{1,2,3}$. The possible configurations of a dimer, indicated by the three ellipses, are labelled by the lattice site and the direction $\eta \in \{1, 2, 3\}$ of the unit vector $\mathbf{e}_\eta$.

resonance terms which can change the dimer configuration on each elementary plaquette consisting of an up- and a down-facing triangle as shown in Fig. 2, as well as potential energy terms for two parallel dimers on a plaquette. The terms which we consider here are sketched in Fig. 3. We define the canonical bosonic (fermionic) creation and annihilation operators $D_{j,\eta}^\dagger$ and $D_{j,\eta}$ ($F_{j,\eta,\sigma}^\dagger$ and $F_{j,\eta,\sigma}$) of a bosonic (fermionic) dimer emanating from lattice site $j$ in one of the three directions $\mathbf{e}_\eta$ with $\eta \in \{1, 2, 3\}$, as depicted in Fig. 2. For fermionic dimers the index $\sigma$ labels the two spin components. In order to make a connection to the electronic Hilbert space of a Hubbard- or t-J model, the dimer operators can be expressed in terms of electron creation and annihilation operators $c_{j\sigma}^\dagger$ and $c_{j\sigma}$ as

$$D_{j,\eta}^\dagger \quad \sim \quad \frac{1}{\sqrt{2}}\left(c_{j\uparrow}^\dagger c_{j+\mathbf{e}_\eta\downarrow}^\dagger - c_{j\downarrow}^\dagger c_{j+\mathbf{e}_\eta\uparrow}^\dagger\right), \tag{1}$$

$$F_{j,\eta,\sigma}^\dagger \quad \sim \quad \frac{1}{\sqrt{2}}\left(c_{j\sigma}^\dagger + c_{j+\mathbf{e}_\eta\sigma}^\dagger\right), \tag{2}$$

up to a phase factor which depends on a gauge choice [20]. Using these dimer operators the Hamiltonian takes the form

$$
\begin{aligned}
H \quad = \quad & -J \sum_j D_{j,1}^\dagger D_{j+\mathbf{e}_2,1}^\dagger D_{j,2} D_{j+\mathbf{e}_1,2} + \ldots \\
& +V \sum_j D_{j,1}^\dagger D_{j+\mathbf{e}_2,1}^\dagger D_{j+\mathbf{e}_2,1} D_{j,1} + \ldots \\
& -t_1 \sum_{j,\sigma} F_{j,1,\sigma}^\dagger D_{j+\mathbf{e}_2,1}^\dagger F_{j+\mathbf{e}_2,1,\sigma} D_{j,1} + \ldots \\
& +v_1 \sum_{j,\sigma} F_{j,1,\sigma}^\dagger D_{j+\mathbf{e}_2,1}^\dagger D_{j+\mathbf{e}_2,1} F_{j,1,\sigma} + \ldots \\
& -t_2 \sum_{j,\sigma} F_{j,1,\sigma}^\dagger D_{j+\mathbf{e}_2,1}^\dagger D_{j,2} F_{j+\mathbf{e}_1,2,\sigma} + \ldots,
\end{aligned}
\tag{3}
$$

where the dots indicate hermitean conjugate terms and analogous terms defined on the other two elementary plaquettes shown in Fig. 2, as well as symmetry related $t_1$, $t_2$ and $v_1$ terms, where bosonic and fermionic dimers are interchanged and/or rotated in the initial or final state. As already mentioned, the density of fermionic dimers is equal to the density of doped holes away from the half-filled electron band. At half-filling, i.e. without any fermionic dimers,

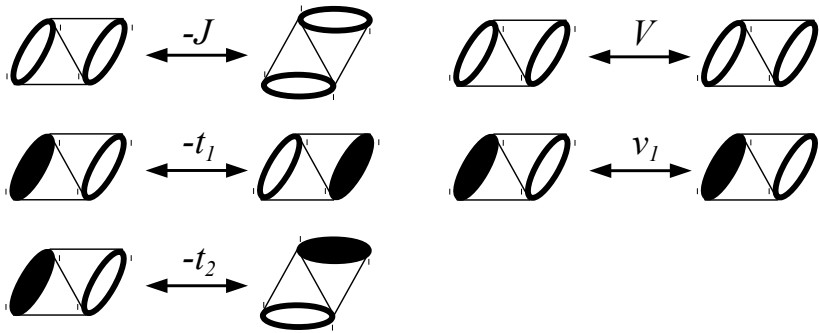

Figure 3: Definition of plaquette terms appearing in our Hamiltonian in Eq. (3) and their respective transition amplitudes. Equivalent terms on the other two elementary plaquettes indicated in Fig. 2 as well as symmetry related terms are not shown.

only the terms in the first two lines of Eq. (3) remain and the Hamiltonian reduces to the Rokhsar-Kivelson model, which is exactly solvable at the so-called RK-point $J = V$. At this special point the Hamiltonian can be written as a sum of projectors on each elementary plaquette and the ground state is an equal weight superposition of all bosonic dimer coverings. It is important to note that the RK-point in the triangular lattice model is part of an extended Z2 spin liquid phase [26]. This is in contrast to the analogous model on the square lattice, where the RK-point is a singular point in the phase diagram exhibiting a $U(1)$ spin liquid ground state.

Since the Hamiltonian (3) is local, the Hilbert space splits into different topological sectors depending on the lattice topology, in precise analogy to the RK-model on the triangular lattice. Using periodic boundary conditions e.g. in x-direction, the parity associated with the number of dimers crossing an arbitrary closed path around the lattice in x-direction, as indicated by the dashed line in Fig. 1, is conserved under arbitrary local operations on the dimer Hilbert space, which thus splits into two topological superselection sectors. This Z2 topological order is an important property of metallic phases realized in this model and allows for a modification of the conventional Luttinger count [27]. As we will see below, the model in Eq. (3) exhibits a metallic ground state where the Fermi volume is proportional to the density of fermionic dimers, i.e. proportional to the density of holes ($p$) away from half-filling. This is in contrast to the conventional Luttinger count for an ordinary Fermi-liquid metal, where the Fermi volume is proportional to the total density of holes measured from the fully filled band (i.e. $1 + p$).

In the following we are going to construct exact ground states for the full model including fermionic dimers. Note that we did not include purely fermionic plaquette resonance terms in Eq. (3), as this model is intended to describe systems at small hole doping slightly below half filling, where the density of fermionic dimers is small and such terms are not expected to play a prominent role.

## 3   Exact ground state solution

We proceed by constructing the exact ground state of a single fermionic dimer interacting with a background of bosonic dimers at a specific line in parameter space. A generalization to an arbitrary number of fermions (also with spin) is then straightforward. Our notation and construction follows closely Ref. [23].

Exact ground states can be constructed for the special choice of parameters $J = V$ and

$\nu_1 = t_2 = -t_1$, in which case the Hamiltonian reduces to a sum of projectors

$$H = H_{\text{RK}} + \nu_1 \sum_{j,\alpha} P_{j,\eta} \, , \tag{4}$$

where $H_{\text{RK}}$ is the standard Rokhsar-Kivelson Hamiltonian (i.e. the first two lines of Eq. (3)) at the special RK-point $J = V$ [21] and we defined the three different plaquette projectors $P_{j,\eta} = |\phi_{j,\eta}\rangle\langle\phi_{j,\eta}|$ with $\eta \in \{1, 2, 3\}$ for every lattice site $j$ and plaquette $\eta$ as defined in Fig. 2. Furthermore,

$$|\phi_{j,1}\rangle = |\langle\!\!\!\diagup\rangle + |\langle\!\!\!\diagdown\rangle - |\langle\!\!\!\rangle - |\langle\!\!\!\rangle \, , \tag{5}$$

$$|\phi_{j,2}\rangle = |\rangle + |\rangle - |\rangle - |\rangle \, , \tag{6}$$

$$|\phi_{j,3}\rangle = |\rangle + |\rangle - |\rangle - |\rangle \, . \tag{7}$$

Here white ellipses denote bosonic spin-singlet dimers and black ellipses represent fermionic dimers. Since we focus on a single fermionic dimer here, the spin index is suppressed. Also note that the Hamiltonian in Eq. (4) is positive semi-definite and thus all states with zero energy are automatically ground states.

We define basis states with one fermionic dimer fixed at a position specified by the lattice site $j$ and the direction $\eta \in \{1, 2, 3\}$ (see Fig. 2 for a definition)

$$|(j,\eta)\rangle = F_{j,\eta}^{\dagger} D_{j,\eta} |RK\rangle \, , \tag{8}$$

where $|RK\rangle$ is the usual Rokhsar-Kivelson wave function, i.e. the equal weight superposition of all hard-core coverings with bosonic spin-singlet dimers in a given topological sector. It is important to note that these basis states are ground states of the RK Hamiltonian $H_{\text{RK}}$ at the exactly solvable RK point $J = V$ per construction, i.e. $H_{\text{RK}}|(j,\eta)\rangle = 0$, because applying the RK Hamiltonian to a plaquette with a fermionic dimer gives zero. Moreover, applying the plaquette projectors $P_{\ell,\alpha}$ to these basis states gives

$$P_{\ell,1}|(j,\eta)\rangle = \left[\delta_{\eta,1}(\delta_{j,\ell} + \delta_{j,\ell+e_2}) - \delta_{\eta,2}(\delta_{j,\ell} + \delta_{j,\ell+e_1})\right]|\phi_{\ell,1}\rangle \, , \tag{9}$$

$$P_{\ell,2}|(j,\eta)\rangle = \left[\delta_{\eta,2}(\delta_{j,\ell} + \delta_{j,\ell+e_3}) - \delta_{\eta,3}(\delta_{j,\ell} + \delta_{j,\ell+e_2})\right]|\phi_{\ell,2}\rangle \, , \tag{10}$$

$$P_{\ell,3}|(j,\eta)\rangle = \left[\delta_{\eta,1}(\delta_{j,\ell} + \delta_{j,\ell+e_3}) - \delta_{\eta,3}(\delta_{j,\ell} + \delta_{j,\ell+e_1})\right]|\phi_{\ell,3}\rangle \, . \tag{11}$$

At the exactly solvable line the ground state turns out to be highly degenerate and different ground states $|\mathbf{p}\rangle$ can be parametrized by a lattice momentum $\mathbf{p}$. We make the ansatz

$$|\mathbf{p}\rangle = \sum_{j,\eta} e^{i\mathbf{p}\cdot\mathbf{R}_j} c_\eta(\mathbf{p})|(j,\eta)\rangle \, , \tag{12}$$

where $\mathbf{R}_j$ is the lattice position of site $j$ and determine the complex coefficients $c_\eta(\mathbf{p})$ by demanding that $H|\mathbf{p}\rangle = 0$. Since $H_{\text{RK}}|\mathbf{p}\rangle = 0$ per construction, this immediately leads to the three equations

$$\sum_{\ell} P_{\ell,\eta}|\mathbf{p}\rangle = 0 \tag{13}$$

for each $\eta \in \{1, 2, 3\}$. Using Eqs. (9), (10), (11) these equations reduce to

$$c_1(\mathbf{p})\left(1 + e^{ip_2}\right) - c_2(\mathbf{p})\left(1 + e^{ip_1}\right) = 0 \, , \tag{14}$$

$$c_2(\mathbf{p})\left(1 + e^{ip_3}\right) - c_3(\mathbf{p})\left(1 + e^{ip_2}\right) = 0 \, , \tag{15}$$

$$c_1(\mathbf{p})\left(1 + e^{ip_3}\right) - c_3(\mathbf{p})\left(1 + e^{ip_1}\right) = 0 \, , \tag{16}$$

where we defined $p_\eta \equiv \mathbf{p} \cdot \mathbf{e}_\eta$. Together with the normalization condition

$$\langle \mathbf{p}|\mathbf{p}\rangle = N Q_c[(j,\eta)] \sum_\eta |c_\eta(\mathbf{p})|^2 = 1 \, , \tag{17}$$

with $N$ the number of lattice sites and $Q_c[(j,\eta)] = \langle (j,\eta)|(j,\eta)\rangle = 1/6$ as the average dimer density per bond, these equations are solved by

$$c_\eta(\mathbf{p}) = \sqrt{\frac{6}{N}} \, \frac{1 + e^{ip_\eta}}{\sqrt{\sum_\zeta |1 + e^{ip_\zeta}|^2}} \, . \tag{18}$$

The above construction can be straightforwardly generalized to an arbitrary number $N_f$ of fermionic dimers, since any of the projectors in the Hamiltonian (4) applied to a plaquette with more than one fermionic dimer gives zero. Accordingly the states

$$|\mathbf{p}_1, \ldots \mathbf{p}_{N_f}\rangle = \sum_{j_1, \eta_1, \ldots j_{N_f}, \eta_{N_f}} e^{i \sum_j \mathbf{p}_j \cdot \mathbf{R}_j} c_{\eta_1}(\mathbf{p}_1) \ldots c_{\eta_{N_f}}(\mathbf{p}_{N_f}) |(j_1, \eta_1), \ldots, (j_{N_f}, \eta_{N_f})\rangle \, , \tag{19}$$

with the coefficients $c_\eta(\mathbf{p})$ given in Eq. (18) are exact, zero-energy ground states of the Hamiltonian (4) with $N_f$ fermionic dimers. Also note that these states are anti-symmetric as required, since the basis states $|(j_1, \eta_1), \ldots, (j_{N_f}, \eta_{N_f})\rangle = F^\dagger_{j_1, \eta_1} D_{j_1, \eta_1} \ldots F^\dagger_{j_{N_f}, \eta_{N_f}} D_{j_{N_f}, \eta_{N_f}} |\text{RK}\rangle$ are anti-symmetric under the exchange of two fermionic dimers.

## 4  Perturbing away from the exactly solvable line: Z2-FL*

The massive ground state degeneracy is immediately lifted away from the exactly solvable line. Within first order perturbation theory the ground states $|\mathbf{p}\rangle$ obtain an energy shift $\varepsilon(\mathbf{p})$, which can be interpreted as a fermion dispersion. Accordingly, at a finite density of fermionic dimers the ground state corresponds to a Fermi sea with the lowest energy states $|\mathbf{p}\rangle$ occupied up to the Fermi energy. This state realizes a fractionalized Fermi liquid with Z2 topological order ( Z2-FL*) and a small Fermi surface, the volume of which is determined by the density of fermionic dimers, i.e. the density of holes ($p$) away from half filling, rather than the total density of holes ($1 + p$) measured from the completely filled band.

Here we compute the dispersion $\varepsilon(\mathbf{p})$ for small deviations of the amplitude $t_1 \to t_1 + \delta t_1$ in the Hamiltonian (4) away from the exactly solvable line $v_1 = t_2 = -t_1$ using first order perturbation theory. A straightforward computation gives

$$
\begin{aligned}
\varepsilon(\mathbf{p}) &= \langle \mathbf{p}|\Delta H|\mathbf{p}\rangle \\
&= -\delta t_1 \sum_{i, \eta_i, j, \eta_j, \ell} e^{-i\mathbf{p} \cdot (\mathbf{R}_i - \mathbf{R}_j)} c^*_{\eta_i}(\mathbf{p}) c_{\eta_j}(\mathbf{p}) \langle (i, \eta_i)|F^\dagger_{\ell,1,\sigma} D^\dagger_{\ell+e_2,1} F_{\ell+e_2,1,\sigma} D_{\ell,1} + \ldots |(j, \eta_j)\rangle \\
&= -\frac{2 \delta t_1}{3} \frac{\epsilon^2_{\mu\nu\lambda}(1 + \cos p_\mu)(\cos p_\nu + \cos p_\lambda)}{\sum_\zeta |1 + e^{ip_\zeta}|^2} \, ,
\end{aligned}
\tag{20}
$$

where $\epsilon_{\mu\nu\lambda}$ is the antisymmetric tensor and sums over repeated greek indices $\mu, \nu, \lambda \in \{1, 2, 3\}$ are implied. In the last line we used that $\langle (\ell, 1)|(\ell + \mathbf{e}_2, 1)\rangle = Q_c[(\ell, 1), (\ell + \mathbf{e}_2, 1)] = 1/18$ is the classical dimer correlation function of two parallel dimers on a plaquette, which can be computed using standard Grassmann techniques [28].

In Fig. 4 we check the validity of the perturbative expansion by comparing Equ. (20) with numerical results. Shown is the dispersion along the high symmetry line $\Gamma - M - K - \Gamma$ for

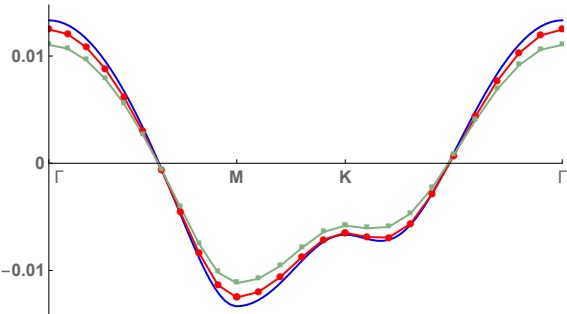

Figure 4: Fermion dispersion relation $\varepsilon(\mathbf{k})$ as function of momentum along the high symmetry line $\Gamma-M-K-\Gamma$ in the vicinity of the exactly solvable line for $\delta t_1 = 0.01$. Blue solid line: perturbative result from Eq. (20), red line with circles: exact diagonalization results on a $4 \times 4$ lattice, green line with squares: exact diagonalization results on a $6 \times 6$ lattice.

$\delta t_1 = -0.01$. The numerical results were obtained using Lanczos exact diagonalization of the Hamiltonian (3), where we computed the ground state energy of a single fermionic dimer with fixed momentum $\mathbf{p}$ interacting with a background of bosonic dimers on lattices of size $4 \times 4$ and $6 \times 6$ with twisted boundary conditions to increase the momentum resolution. The numerical results show reasonably good agreement with the perturbative result in Eq. (20) and the small discrepancies can be attributed to finite size effects.

Fig. 5 shows a contour plot of the perturbative dispersion $\varepsilon(\mathbf{p})$ for $\delta t_1 = -0.25$. Note that the dispersion minima are located at the $M$ points on the edges of the Brillouin zone. This feature persists away from the exactly solvable line and is a generic feature of the fermion dispersion for realistic parameter choices. Indeed, the dimer resonance amplitudes can be estimated by computing matrix elements of a simple nearest-neighbor tight binding Hamiltonian $H_t = -t \sum_{\langle i,j \rangle} c_i^\dagger c_j$ of electrons on the triangular lattice of the form

$$t_1 \simeq -\langle \text{⬭}|H_t|\text{⬭}\rangle = -\frac{3t}{4} \, , \tag{21}$$

$$t_2 \simeq -\langle \text{⬭}|H_t|\text{⬭}\rangle = \frac{t}{2} \, , \tag{22}$$

where $t$ is the nearest-neighbor electron hopping amplitude. Note in particular that $t_1$ is negative and 50% larger in magnitude than $t_2$. This puts the parameters in the vicinity of the exactly solvable line, where $t_1 = -t_2$. Accordingly we might suspect that the exact solution captures physical properties in the realistic parameter regime. In Fig. 6 we show a plot of the numerically obtained fermion dispersion for the choice of parameters $J = V = 1$, $t_1 = -3$, $t_2 = 2$ and $v_1 = 2$, corresponding to a nearest neighbor electron hopping amplitude $t = 4J$. Again, the data was obtained using Lanczos exact diagonalization of the Hamiltonian (3) for a single fermionic dimer with fixed momentum on a lattice of size $4 \times 4$ with twisted boundary conditions. Note that the minima of the fermion dispersion remain at the $M$ points for this choice of parameters.

It is also important to note that the Fermi surface of the fermionic dimers as obtained from the dispersion $\varepsilon(\mathbf{p})$ in Eq. (20) has a direct imprint on the electronic Fermi surface. The argument is analogous to Ref. [23] and here we focus on the main idea. Defining the operators

$$f_{\mathbf{p},\sigma}^\dagger = \sum_{j,\eta} e^{i\mathbf{p}\cdot\mathbf{R}_j} c_\eta(\mathbf{p}) F_{j,\eta,\sigma}^\dagger D_{j,\eta} \tag{23}$$

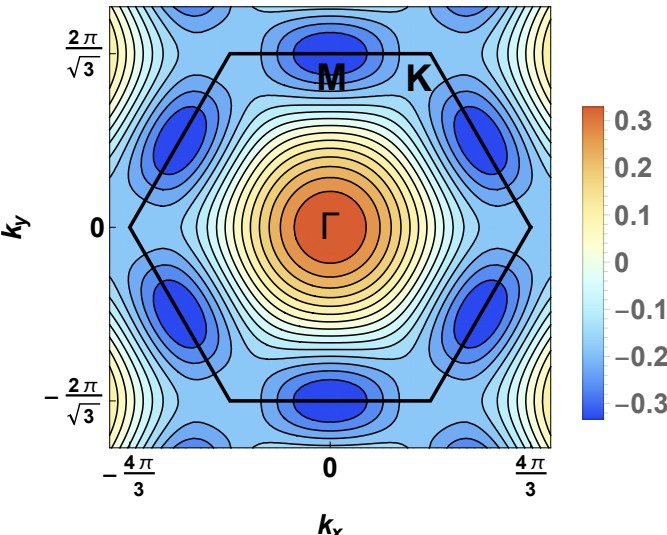

Figure 5: Plot of the perturbative result for the fermion dispersion relation $\varepsilon(\mathbf{k})$ from Eq. (20) for $\delta t_1 = -0.25$.

and acting with them on the bosonic RK ground state $|RK\rangle$ generates the exact ground states in Equ. (19). Note that we've reintroduced the spin label here. The operators $f^\dagger_{\mathbf{p},\sigma}$ and $f_{\mathbf{p},\sigma}$ obey canonical anti-commutation relations $\{f_{\mathbf{p},\sigma}, f^\dagger_{\mathbf{q},\sigma'}\} = \delta_{\mathbf{p},\mathbf{q}}\delta_{\sigma,\sigma'}$ in the thermodynamic limit on the Hilbert space spanned by the states in Equ. (19). This can be shown by computing $\left|\{f_{\mathbf{p},\sigma}, f^\dagger_{\mathbf{q},\sigma'}\}|RK\rangle\right|^2$ and realizing that it is proportional to the Fourier transform of the classical dimer correlation function. Within first order perturbation theory the Hamiltonian thus takes the form of a free fermion Hamiltonian $H = \sum_{\mathbf{p},\sigma} \varepsilon(\mathbf{p}) f^\dagger_{\mathbf{p},\sigma} f_{\mathbf{p},\sigma}$ in the vicinity of the exactly solvable line.

Lastly, the operators $f^\dagger_{\mathbf{p},\sigma}$ can be directly related to electron annihilation operators $c_{\mathbf{p},\sigma}$, which can be seen as follows. On the dimer Hilbert space the electron destruction operator $c_{j,\sigma}$ on lattice site $j$ can be uniquely written in terms of the dimer operators in Eqs. (1), (2) as [20]

$$c_{j,\sigma} = \frac{\epsilon_{\sigma\sigma'}}{2} \sum_\eta \left( F^\dagger_{j,\eta,\sigma'} D_{j,\eta} + F^\dagger_{j-\mathbf{e}_\eta,\eta,\sigma'} D_{j-\mathbf{e}_\eta,\eta} \right) , \qquad (24)$$

which can be checked by computing matrix elements on both sides of the equation in the dimer Hilbert space. Fourier transforming this relation it is straightforward to show that

$$c_{\mathbf{p},\sigma} = \epsilon_{\sigma,\sigma'} \sqrt{Z_\mathbf{p}} f^\dagger_{-\mathbf{p},\sigma'} , \qquad (25)$$

where $Z_\mathbf{p} = \sum_\eta |1 + e^{-ip_\eta}|^2/24$ is the electronic quasiparticle residue. In the vicinity of the exactly solvable line the electron spectral function thus takes the form

$$A_e(\mathbf{p}, \omega) = Z_\mathbf{p} \delta(\omega + \varepsilon(\mathbf{p}) - \mu) , \qquad (26)$$

where we introduced the chemical potential $\mu$ to fix the average density of fermionic dimers. This demonstrates that the small Fermi surface of fermionic dimers is directly imprinted on the electronic Fermi surface and proves that the ground state is indeed a fractionalized Fermi liquid with Z2 topological order. It features a small Fermi surface with an enclosed volume proportional to the density of doped holes away from half filling in the absence of any broken symmetries.

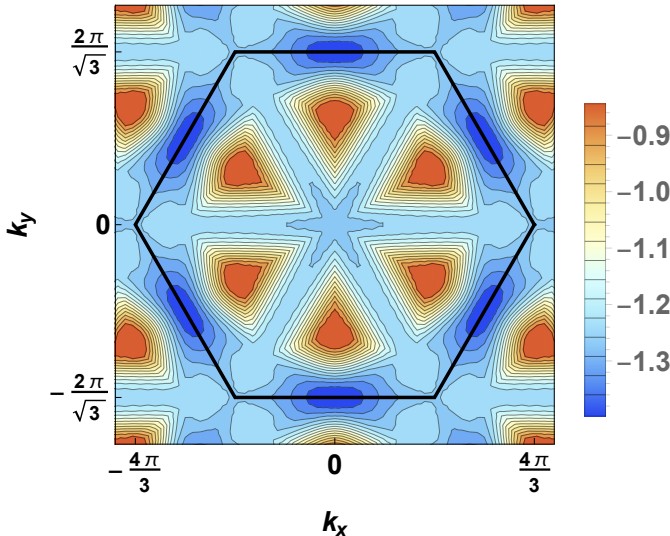

Figure 6: Exact diagonalization result for the dispersion of a single fermionic dimer interacting with a background of bosonic dimers on a lattice of 4×4 sites with twisted boundary conditions. Parameters: $J = V = 1$, $t_1 = -3$, $t_2 = 2$ and $v_1 = 2$.

## 5 Application to Twisted Bilayer Graphene

In this section we comment on a possible application of the previously introduced triangular lattice quantum dimer model in the context of magic-angle twisted bilayer graphene (TBG). Recent experiments on TBG showed correlated Mott-like insulating behavior at half filling of the lower Moiré mini-band near the charge neutrality point [6, 29, 30]. Moreover, TBG becomes superconducting upon doping the correlated insulator with electons or holes. Interestingly, Shubnikov - de Haas as well as Hall resistivity measurements above the superconducting transition temperature on the hole-doped side of the correlated insulator indicate an anomalously small charge carrier density, which is equal to the density of doped carriers measured from the correlated insulator at half filling [6]. This is in remarkable analogy to Hall measurements in the pseudogap phase of underdoped cuprates [5]. One possible explanation of a small carrier density involves translational symmetry breaking orders, which enlarge the size of the unit cell and reconstruct the Fermi surface into small pockets. A different possibility is the presence of topological order, which can give rise to small pocket Fermi surfaces and a small carrier density as well, as discussed in this work. So far no symmetry breaking orders have been observed in TBG which could explain the small carrier density, consequently the Z2-FL* scenario discussed here could provide a viable explanation.

The microscopic details of TBG are rather complex and a lot of effort has been put into developing realistic tight binding models [31–34]. In the following we argue that the triangular lattice dimer model studied here can be viewed as a very simplistic toy model to gain intuition about the unconventional metallic state near the Mott-like insulator at half filling of the lower Moiré mini-band. Indeed, electric charge in TBG is localized at the AA stacked regions of the two twisted graphene layers, which form a triangular lattice [35]. Even though symmetry constraints preclude the definition of a triangular lattice tight binding model for TBG and microscopic models have a lower rotational symmetry, it is expected that the six-fold rotational symmetry of the triangular lattice emerges in effective low-energy theories. As such, triangular lattice models like the one discussed here should be able to capture effects of local correlations on the low-energy properties of TBG [36].

An additional important complication is the two-fold valley degeneracy in TBG. In our simplistic picture this would lead to two degenerate orbitals per triangular lattice site due to the valley degree of freedom, i.e. a four fold degeneracy per lattice site in total, if spin is included. This fourfold degeneracy is indeed observed in the Landau level structure emanating from the charge neutrality point [6]. However, the Landau level degeneracy in the metallic state emanating from the Mott-like insulator at half filling is only two-fold. It has been argued that this could be due to the presence of so-called inter-valley coherent order, where the valley quantum number is lost [33]. The spin quantum number is likely to remain unaffected, since the superconductor, which develops out of this unconventional metallic state, can be suppressed by a parallel magnetic field and thus spin-singlet pairing is probable. Using this scenario as a basis, our triangular quantum dimer model can be viewed as a simple toy-model for the remaining two-fold degenerate spin-states per triangular lattice site in a conjectured inter-valley coherent phase and offers a possible explanation for the low-charge carrier density upon hole-doping the Mott-like insulator. As discussed in the main part of this manuscript, a prominent signature of the Z2-FL* state are the small hole-pockets centered at the M points of the Moiré mini-Brillouin zone in the absence of translational symmetry breaking orders. However, the Fermi-pockets at the three distinct M points should give rise to an additional three-fold degeneracy of the Landau-levels, which is not observed in experiments. This could be due to two reasons: either the Fermi surface undergoes an additional reconstruction in the presence of a strong magnetic field, or our simple estimate of the dimer hopping amplitudes from a single-band, nearest-neighbor triangular tight-binding model does not hold for TBG and the dispersion minimum is in fact at the Γ point. In any case, the location of the small Fermi pockets in TBG could be determined in principle using either ARPES measurements with very high energy and momentum resolution, which is currently out of reach, or using quasiparticle interference in STM experiments. We also note here that a different route towards a description of metallic states with Z2 topological order in TBG is discussed in Ref. [36].

# 6    Discussion and Conclusions

We presented a generalized, two-species quantum dimer model on the triangular lattice which features a fractionalized Fermi liquid ground state with Z2 topological order (Z2-FL*). An exact ground state solution was constructed for a specific line in parameter space. At this exactly solvable line the ground state is highly degenerate and can be interpreted as a flat fermionic band. Using perturbation theory we computed the fermion dispersion away from the exactly solvable line and showed that the ground state is indeed a Z2-FL* with a modified Luttinger count, where the electronic Fermi surface encloses a volume proportional to the density of doped holes away from half filling. In particular we showed that the Fermi surface consists of small hole pockets in the vicinity of the M points on the edges of the Brillouin zone at low doping. This feature is quite robust for realistic parameter choices derived from a single-band tight-binding model of electrons on a triangular lattice and can be used as a signature of the Z2-FL* state on the triangular lattice.

Even though we focused on the unconventional metallic Z2-FL* state in this work, the complete phase diagram of this model also contains symmetry broken states that we did not consider here. In the undoped case, transitions to symmetry broken valence bond solid (VBS) phases can be described as confinement transitions of the effective Z2 gauge theory for the resonating valence bond phase [37,38]. It is an interesting open problem how such confinement transitions are affected by the presence of a finite density of fermionic dimers. Related questions on the square lattice have been studied recently using sign-problem free quantum Monte-Carlo [39,40] and it would be interesting to extend this approach to the model studied

here.

## Acknowledgements

We are grateful to Johannes Feldmeier and Sebastian Huber for helpful discussions.

**Funding information** This research was supported by the German Excellence Initiative via the Nano Initiative Munich (NIM) as well as by the Deutsche Forschungsgemeinschaft (DFG, German Research Foundation) under Germany's Excellence Strategy – EXC-2111 – 390814868.

## A    Kagome lattice dimer model

The construction presented in Sec. 3 can be straightforwardly generalized to different lattice geometries, such as the kagome lattice, which we will briefly discuss in this appendix. A general dimer Hamiltonian again consists of various plaquette resonance terms, in analogy to Eq. 3 on the triangular lattice. We consider terms on the four different plaquettes shown in Fig. 7, namely the elementary hexagon, as well as the three diamond shaped loops of length eight. Instead of specifying the various resonance terms in the Hamiltonian in detail, we directly define an exactly solvable Hamiltonian in terms of plaquette projectors, as in Eq. (4). Again, we keep an RK-like term $H_{\mathrm{RK}}$, which ensures that the ground state without fermionic dimers is an equal weight superposition of all bosonic dimer coverings. Moreover, we define the four plaquette projectors $P_{j,\eta} = |\phi_{j,\eta}\rangle\langle\phi_{j,\eta}|$ involving a fermionic dimer, where

$$|\phi_{j,1}\rangle = |\hexagon\rangle + |\hexagon\rangle + |\hexagon\rangle + |\hexagon\rangle + |\hexagon\rangle + |\hexagon\rangle , \tag{27}$$

$$|\phi_{j,2}\rangle = |\diamond\rangle + |\diamond\rangle + |\diamond\rangle + |\diamond\rangle + |\diamond\rangle + |\diamond\rangle + |\diamond\rangle + |\diamond\rangle \tag{28}$$

and analogous terms $|\phi_{j,3}\rangle$ and $|\phi_{j,4}\rangle$ for the other two diamond shaped plaquettes. Note that we have six independent bonds per kagome unit cell (see Fig. 7) and thus we need to determine six coefficients $c_{\eta}(\mathbf{p})$ with $\eta \in \{1,\ldots,6\}$ in our ansatz for the ground state wave function in Eq. (12). Consequently we would need five linearly independent plaquette projectors together with the normalization condition in order to uniquely specify the coefficients $c_{\eta}(\mathbf{p})$. With our four plaquette operators the coefficients are underdetermined, but we can define exact ground states nonetheless.

Applying the four projectors to our ansatz (12) and requiring that the eigenenergy is zero leads to the four equations

$$c_1(\mathbf{p}) + c_2(\mathbf{p}) + c_3(\mathbf{p}) + c_4(\mathbf{p}) + c_5(\mathbf{p}) + c_6(\mathbf{p}) = 0 , \tag{29}$$

$$c_1(\mathbf{p})\left(1 + e^{i2p_1}\right) + c_2(\mathbf{p})\left(1 + e^{i2p_3}\right) + c_4(\mathbf{p})\left(1 + e^{-i2p_1}\right) + c_5(\mathbf{p})\left(1 + e^{-i2p_3}\right) = 0 , \tag{30}$$

$$c_2(\mathbf{p})\left(1 + e^{-i2p_2}\right) + c_3(\mathbf{p})\left(1 + e^{-i2p_1}\right) + c_5(\mathbf{p})\left(1 + e^{i2p_2}\right) + c_6(\mathbf{p})\left(1 + e^{i2p_1}\right) = 0 , \tag{31}$$

$$c_1(\mathbf{p})\left(1 + e^{-i2p_2}\right) + c_3(\mathbf{p})\left(1 + e^{i2p_3}\right) + c_4(\mathbf{p})\left(1 + e^{i2p_2}\right) + c_6(\mathbf{p})\left(1 + e^{-i2p_3}\right) = 0 . \tag{32}$$

Solving these equations for $c_{2,3,5,6}$ together with the normalization condition Eq. (17), where the mean dimer density per bond on the kagome lattice is $Q_c[(i,\eta)] = 1/6$, and setting



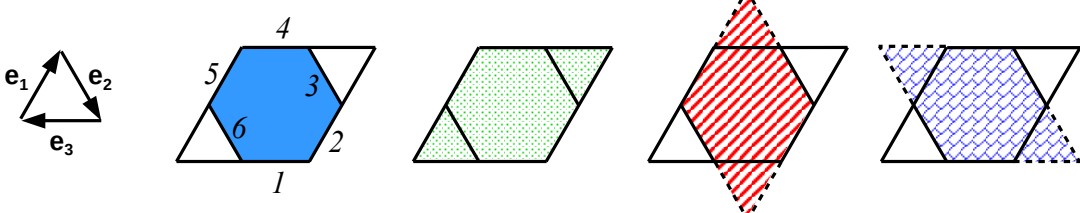

Figure 7: Kagome lattice unit cell with independent bonds enumerated from one to six (left). We consider four different plaquette terms: the elementary hexagon (marked in blue), as well as the three diamond shaped loops of length 8 (marked by green dots, red stripes and blue bricks).

$c_4 = -c_1$ gives a particularly simple solution. We finally obtain

$$c_1(\mathbf{p}) = -c_4(\mathbf{p}) = \sqrt{\frac{6}{N}} \frac{|\sin(2p_3)|}{\sqrt{3 - \cos(4p_1) - \cos(4p_2) - \cos(4p_3)}} \,, \tag{33}$$

$$c_2(\mathbf{p}) = -c_5(\mathbf{p}) = -c_1(\mathbf{p}) \frac{e^{i2(p_1-p_3)}\left(1 - e^{-i4p_1}\right)}{1 - e^{-i4p_3}} \,, \tag{34}$$

$$c_3(\mathbf{p}) = -c_6(\mathbf{p}) = c_1(\mathbf{p}) \frac{e^{i2(p_2-p_3)}\left(1 - e^{-i4p_2}\right)}{1 - e^{-i4p_3}} \,. \tag{35}$$

Again, this solution can be straightforwardly extended to an arbitrary number of fermionic dimers.

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
