# Peer review of "Solvable lattice models for metals with Z2 topological order"

_SciPost Physics, doi:SciPost Phys. 7, 074 (2019)_

## Round 1 · Referee Report · Anonymous · 2019-10-2

Strengths

1. New and exact results on a strongly-interacting topologically ordered model of fermions.
2. Concise and clear presentation of the results.

Weaknesses

1. The physical discussion is restricted to a small range of physical parameters and phases.
2. Connection to twisted angle bilayer graphene physics is not sufficiently substantiated.

Report

The authors study a lattice model of spin-singlets coupled to fermionic dimers, defined on the triangular lattice. Building on a previous study concerning a similar model on the square lattice, an exactly solvable point in parameter space is identified, and its physical properties are thoroughly analyzed. The resulting many-body state, known as a fractionalized Fermi-Liquid, violates Luttinger's theorem through the appearance of Z2 topological order. The exactly solvable point is massively degenerate due to its flat-band spectrum. Perturbing away from this point lifts the degeneracy, giving rise to a non-trivial dispersion. A crucial feature of the triangular lattice model, which distinguishes it from all previous studies, is the stability of a Z2 deconfined phase. This property is in sharp contrast to the finely-tuned RK point on the square lattice. The authors make some connections to recent experimental observations in twisted bilayer graphene. In the appendix, results for a similar model defined on the Kagome lattice are presented.

The paper is very well written and contains a concise yet self-contained presentation of the main results. Identifying an exactly solvable model exhibiting a topologically ordered metallic state is a fundamental topic with direct links to experiments in strongly correlated metals. Therefore, I recommend publication in SciPost.

Below, I list a few points which and I would like the authors to consider.

Requested changes

1. In contrast to the square lattice case that without fine-tuning is always confining, on a triangular lattice, a confinement-deconfinement transition is allowed. Could the authors comment on the nature of such a transition? In particular, its effect on the fermionic degrees of freedom.

2. Could the authors provide a physical understanding of the electronic dispersion? In particular, could the band minimum at the M point be explained through symmetry-based arguments?

3. I find that the discussion on possible connections to twisted to bi-layer graphene physics is slightly unclear. While I understand that inevitably such a link is heuristic, it might be worth devoting a full section to make the discussion more precise.

---

## Round 2 · Author Response

We thank the referee for the very positive report and the helpful comments on our manuscript. Below we give a detailed response to the referee’s questions and comments. We updated our manuscript accordingly and hope this facilitates a timely publication of our work.

With many thanks and best regards, Brin Verheijden, Yuhao Zhao, Matthias Punk

Response to referee’s requested changes:

System Message: WARNING/2 (<string>, line 6)

Title overline too short.

=================================
Response to referee’s requested changes:
=================================

1.) So far we haven’t studied confinement transitions in our dimer model, but this is indeed a very interesting question. Such transitions would be manifest in the spontaneous breaking of lattice symmetries, leading to valence bond solid order for the purely bosonic RK model. In the presence of fermionic dimers we would expect a small Fermi surface which satisfies the conventional Luttinger count, since the Fermi surface is reconstructed due to the breaking of lattice symmetries. Unfortunatley we are not aware of definitive statements about the nature of confinement transitions between the Z2 fractionalized Fermi liquid studied in our work, and a confining phase in terms of an ordinary Fermi liquid with broken symmetries. However, related questions have been investigated recently in several numerical works, which studied square lattice models of fermionic matter coupled to Z2 gauge fields. We’ve added a corresponding comment in the discussion and conclusions section of our revised manuscript.

2.) We are not aware of a symmetry based argument which would explain why the dispersion minimum is at the M points of the Brillouin zone. Note, however, that a change of the sign of the dimer resonance amplitude delta t_1 shifts the position of the dispersion minimum back to the Gamma point. Moreover, a perturbation of the amplitude t_2 from the exactly solvable line (which we didn’t compute here, because we identified t_1 as the important perturbation) can shift the position of the dispersion minimum to the K points at the Brillouin zone corners. The position of the dispersion minima thus clearly depends on microscopic details.

3.) We want to emphasize that our model is at best a very simplistic toy model for the unconventional metallic state that has been observed in magic-angle twisted bilayer graphene (TBG) on the hole-doped side of the Mott-like insulator at a filling of nu = -2. Note that microscopic details of TBG are rather complex, as evidenced by the subtleties encountered in previous works that tried to construct a faithful tight-binding description. For this reason we wanted to refrain from making strong claims about the applicability of our model to TBG, besides pointing a few basic observations. As suggested by the referee, we now discuss relations to TBG in a new section of our manuscript, which contains an extended discussion of what was previously found in the conclusions section.

---

## Round 2 · List of Changes

1.) New section 5 with an extended discussion of the relation of our results to TBG

2.) A new comment on confinement transitions in the Conclusions & Discussions section

3.) new references 34, 37, 38, 39, 40

---

## Editorial Decision

published